# Numerical Study of the Collapse of Multiple Bubbles and the Energy Conversion during Bubble Collapse

**Jing Zhang** [1,2,3], **Lingxin Zhang** [1,2,3,*] and **Jian Deng** [1,2,3]

1    State Key Laboratory of Fluid Power and Mechatronic Systems, Zhejiang University,
     Hangzhou 310027, China; zj121314@yeah.net (J.Z.); zjudengjian@zju.edu.cn (J.D.)
2    Key Laboratory of Soft Machines and Smart Devices of Zhejiang Province, Zhejiang University,
     Hangzhou 310027, China
3    Department of Mechanics, Zhejiang University, Hangzhou 310027, China
*    Correspondence: zhanglingxin@zju.edu.cn; Tel.: +86-136-5671-5841

**Abstract:** This paper investigates numerically the collapses of both a single cavitation bubble and a cluster consisting of 8 bubbles, concerning mainly on the conversions between different forms of energy. Direct numerical simulation (DNS) with volume of fluid (VOF) method is applied, considering the detailed resolution of the vapor-liquid interfaces. First, for a single bubble near a solid wall, we find that the peak value of the wave energy, or equivalently the energy conversion rate decreases when the distance between the bubble and the wall is reduced. However, for the collapses of multiple bubbles, this relationship between the bubble-wall distance and the conversion rate reverses, implying a distinct physical mechanism. The evolutions of individual bubbles during the collapses of multiple bubbles are examined. We observe that when the bubbles are placed far away from the solid wall, the jetting flows induced by all bubbles point towards the cluster centre, while the focal point shifts towards the solid wall when the cluster is very close to the wall. We note that it is very challenging to consider thermal and acoustic damping mechanisms in the current numerical methods, which might be significant contributions to the energy budget, and we leave it open to the future studies.

**Keywords:** cavitation bubble, multiple bubble collapse, pressure wave energy, energy conversion rate

## 1. Introduction

As a natural phenomenon, cavitation occurs frequently in a variety of engineering applications. As the local pressure in liquid is lower than its saturated vapor pressure, gas nucleus can develop into cavitation bubbles. When the local pressure is recovered or driven by high pressure waves, the cavitation bubbles collapse, leading to a series of physical scenarios such as micro jetting flows, shock wave emission, heat release etc. In hydraulic systems, cavitation can cause material damage, loss of power, and induce noises. On the other hand, bubble collapse can also be utilized as a tool in the fields of medical treatments and mining, such as drug delivery, ultrasound operation and cavity jet exploits [1]. Numerous experimental and numerical studies have been performed to study the bubble dynamics and the collapses of cavitation bubbles, in order to control or reinforce the effects of cavitation bubble collapse. However, the dynamics of multiple cavitation bubbles and their collapses, particularly the underlying behavior of energy conversion have not been well understood.

Historically, studies on bubble dynamics can date back to 1917, when Reyleigh proposed a theoretical spherical-bubble model [2]. This model was for an ideal spherical bubble, neglecting the effects of viscosity, surface tension, compressibility and mass transfer. It was followed by a series of modifications and perfections [3–8].

In the past few decades, a large amount of works have been undertaken to understand the characteristics of bubble collapse [9–24]. Lauterborn recorded the process of a single bubble collapsing near a rigid wall, in which the shock wave emission was captured by a schlieren system [25–27]. Moreover, the particle image velocimetry (PIV) technique was adopted to monitor the velocity fields by lauterborn & Vogel [28], who captured the micro jetting flows as the bubble collapses asymmetrically. They revealed the relationship between the noises induced by the bubble collapse and the dimensionless distance $\gamma$ ($\gamma = L/R_{max}$, where $L$ is the distance between the bubble centre and the rigid wall, and $R_{max}$ is the maximum bubble radius). Gonzalez-Avila measured the pressure emission of laser-induced bubble collapse [29], in which the pressure amplitude was found to be up to $1k$ Bar when the bubble was close to the rigid boundary [30]. Merouan [31] measured the temperature variations during the growth and collapse of a single cavitation bubble. More recently, Fortes et al. proposed an analytic model to examine the pressure wave emission during bubble collapse, with the conversion rate of the wave energy found to be strongly influenced by gas content in the bubble [32].

However, not many efforts have been made to study the pressure wave emission from the collapses of multiple bubbles due to the difficulties in experimental setup and the consideration of compressibility of liquid in numerics [33–36]. Generally, a bubble cluster collapses inward, generating a high level impulsive pressure wave, and the symmetries for individual bubbles are lost due to the presence of the rigid wall and their nearby bubbles [37,38]. It has been shown that the shock wave emission focuses inward during the bubble cloud collapse, leading to a very high peak pressure, which can exceed that of the collapse of a single bubble with the equivalent volume [39]. Tiwari et al. [40] simulated the expansion and subsequent collapse of a hemispherical cluster of 50 bubbles adjacent to a plane rigid wall, with the detailed compressible-fluid mechanics of bubble-bubble interactions considered. They found that the peak pressure was associated with the centremost bubble, which caused a corresponding peak pressure on the nearby wall. Despite these advances in the few existing researches, the detailed bubble-scale dynamics of a cluster collapse remain poorly understood.

Here, we investigate the characteristics of the collapses of multiple bubbles by solving directly the Navier-Stokes equations considering the compressiblility of the fluid, with particular concerns on the energy conversion. The volume of fluid method is involved to capture the interfaces between the liquid and the cavitation bubbles. The simulation results are organized into two parts: in the first part, a single bubble is studied as a reference to our further multiple-bubble studies. In the second part, we focus the collapses of multiple bubbles, presenting the unique dynamic behaviors in pressure wave propagation and energy conversion, in contrast to the single bubble situation.

## 2. Numerical Methods

The numerical work is performed using a finite-volume based code. Assuming that the liquid is compressible and the cavitation bubbles are filled with pure vapour, and considering viscosity and surface tension, the governing equations are formularized as the following

$$\frac{d\rho}{dt} + \rho \nabla \overrightarrow{U} = 0, \tag{1}$$

$$\rho \frac{\partial \overrightarrow{U}}{\partial t} + \rho \overrightarrow{U} \cdot \nabla \overrightarrow{U} = \rho \overrightarrow{g} - \nabla p + 2\nabla \cdot \left( \mu \overline{\overline{D}} \right) - \frac{2}{3} \nabla \left( \mu \nabla \cdot \overrightarrow{U} \right) + \sigma \kappa \overrightarrow{N}, \tag{2}$$

$$\frac{d\alpha}{dt} = \frac{\partial \alpha}{\partial t} + \overrightarrow{U} \cdot \nabla \alpha = 0, \tag{3}$$

where $k$ is surface curvature, $\sigma$ is the surface tension coefficient, $\overline{\overline{D}}$ is the strain rate tensor, $\overrightarrow{N}$ is the unit normal vector of the interface, $\alpha$ is the volume fraction of the liquid, and $\rho$ and $\mu$ are the density and the viscosity respectively of the mixture fluid, which are obtained by weighting of the volume fractions:

$$\rho = \alpha \rho_1 + (1 - \alpha)\rho_2, \tag{4}$$

$$\mu = \alpha\mu_1 + (1 - \alpha)\mu_2, \tag{5}$$

where subscripts 1 and 2 denote liquid phase (water) and gas phase (vapour), respectively. The densities of each phase are calculated by

$$\rho_1 = \rho_{10} + \psi_1 p, \tag{6}$$

$$\rho_2 = \rho_{20} + \psi_2 p, \tag{7}$$

where $\psi_1$ and $\psi_2$ are two constants associated with the respective compressibilities of water and vapour.

In our simulations, the space discretizations are second-order upwind for the convection terms and central differences for the Laplacian terms, respectively. The time discretization is first-order implicit Euler. The pressure-velocity coupling is obtained using the Pressure Implicit Split Operator (PISO) scheme. The preconditioned conjugate gradient (PCG) method is used to treat the pressure equation and the preconditioned biconjugate gradient (PBiCG) method is used for the velocity equations.

The computational domain is a cylinder with a radius of 25 mm, and a height of 50 mm. The grid distributions on some specific sections are shown in Figure 1. The bottom of the cylindrical flow domain is set as a rigid wall, the others are far field boundaries. The initial pressure of the liquid phase is 101,325 Pa, and the pressure of vapor is 3154 Pa. The initial flow is quiescent, and the initial bubbles are spherical, with the radius of 2 mm. The geometrical setup of the multiple-bubble case is shown in Figure 2. The dimensionless wall distance is defined as $\gamma = L/R_{max}$. After carefully carrying out self-consistency tests, we find that the current grid with a cell number of 25,581,192, and an initial maximum time step size of $dt = 1 \times 10^{-7}$ s, are sufficient to assure satisfactory independence of the results with respect to both mesh and time discretizations. We note that the time step is adjustable during the simulations to meet the requirement of local courant number $Co = 0.35$.

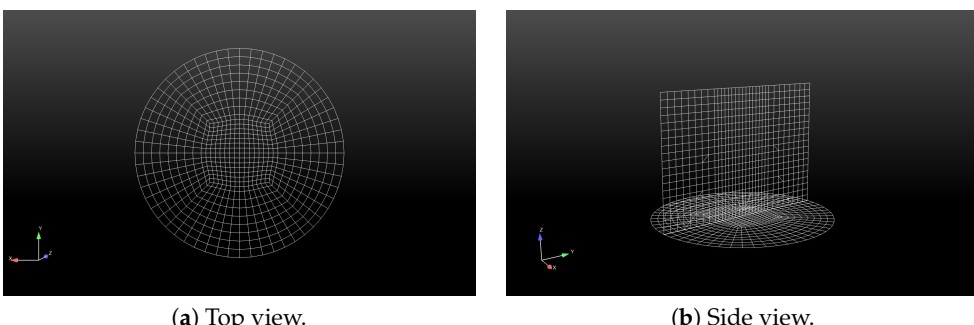

(**a**) Top view.　　　　　　　　　　　　　　(**b**) Side view.

**Figure 1.** (**a**) Top view and (**b**) side view of the grid distributions within the computational domain. Note that the grid in the central area of the cylindrical domain is refined where the bubbles are placed, and only a coarse grid with 10% of the cells is displayed to diagram the grid distributions.

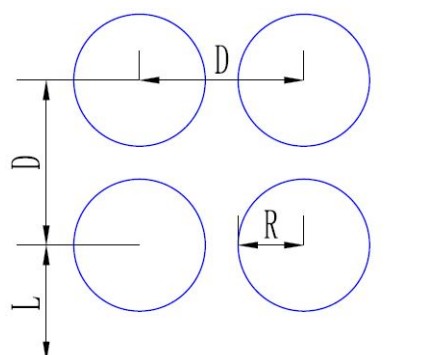

**Figure 2.** Geometrical description for the simulations of multiple bubbles.

To further validate the spatial resolutions, we carry out simulations with different numbers of cells along the diameter of a single bubble. Figure 3 presents the time histories of the dimensionless

radius $R^*$, where $R^* = R/R_{max}$, and the time is non-dimensionalized according to $T^* = t/t_c$. Here, $R_{max}$ is the maximum or the initial radius of the bubble, and $t_c$ is the collapsing time predicated from the Reyleigh-plesset equation. It is observed that the result converges at 34 cells, which accords well with the theoretical solution. We show in Figure 4a the evolutions of bubble shapes during collapse simulated based on the current resolution of 34 cells. The numerical results are consistent with the previous experimental observation [41].

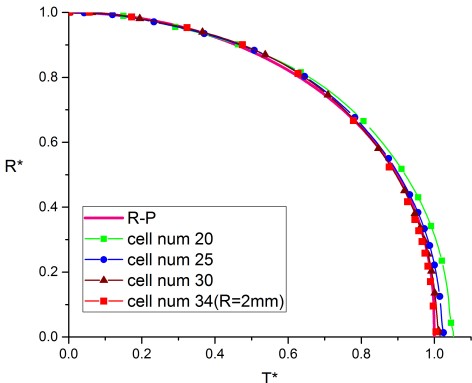

**Figure 3.** Time histories of radius for a single bubble resolved by different number of cells distributed along the diameter. The reference curve is the theoretical solution of the Rayleigh-Plesset equation.

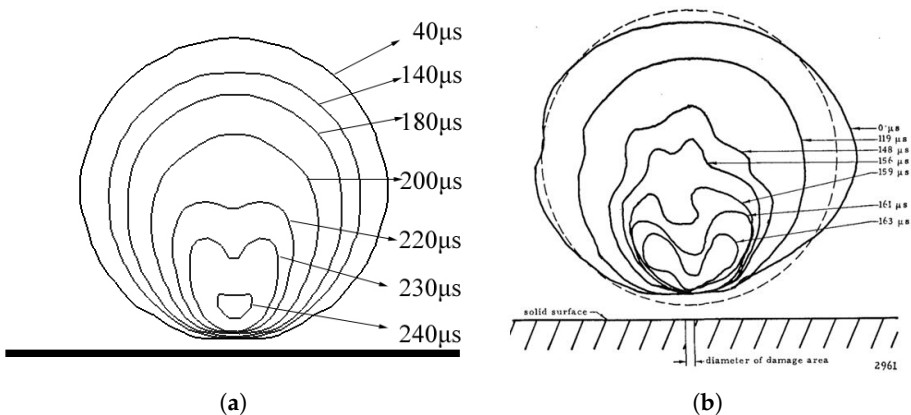

(**a**)                                    (**b**)

**Figure 4.** Variations of the bubble shapes during collapse obtained by (**a**) the present numerical simulations and (**b**) previous experiments [41].

In the following sections, according to [42], the potential energy of a bubble is defined as

$$E_{pot} = \frac{4}{3}\pi R^3 \Delta p, \tag{8}$$

where $R$ is the bubble radius and $\Delta p$ is the pressure difference between two sides of the bubble interface, then the maximum potential energy of a cavitation bubble is defined as $E_{pot-max}$. The total kinetic energy of the flow domain is the integral:

$$E_k = \int \frac{1}{2}\rho U^2 dV. \tag{9}$$

We also define the wave energy emitted by the bubble collapse, following an acoustic approach [28,43], with the following form:

$$E_{wave} = \int \frac{\Delta P^2}{(\rho c)^2} dV, \tag{10}$$

where $c$ is the sound speed in water, $c = 1500$ m/s in the current simulations. The conversion rate of the wave generation is defined as

$$\eta = \frac{E_{wave}}{E_{pot-max}} \tag{11}$$

## 3. Result

### 3.1. Collapse of a Single Bubble

In this section, we study a single cavitation bubble collapsing near a solid wall. To understand the presence of the solid wall on the non-spherical deformation, and consequently the energy conversion, we carry out a series of simulations by varying the distance between the bubble and the solid wall. The dimensionless distance $\gamma$ ranges from 1.5 to 4.0.

In Table 1, we present the simulation results. First, we observe that the energy conversion rate reaches its maximum of 29.29% at $\gamma = 12.5$, which is not difficult to understand since the $\gamma = 12.5$ case corresponds to a distance very far from the solid wall, which we believe that the wall effect can be reasonably neglected. The values of maximum kinetic energy for all cases are around 4.0 mJ, with slight variations among different distances. We note that the energies and the conversions at $\gamma = 2.5$ and $\gamma = 4.0$ are very close, exhibiting nontrivial bounding effects of the solid wall.

**Table 1.** Energies and their conversion rates for various distances for the collapse of a single bubble (note the different units for the two energies).

| $\gamma$ | $E_k$ (mJ) | $E_{wave}$ (μJ) | $\eta$ (%) |
|---|---|---|---|
| 1.5 | 3.94 | 352.2 | 10.73 |
| 1.65 | 3.90 | 418.1 | 12.74 |
| 1.85 | 3.97 | 543.8 | 16.57 |
| 2.0 | 3.94 | 564.2 | 17.19 |
| 2.5 | 4.08 | 608.8 | 18.55 |
| 3.0 | 3.95 | 579.3 | 17.65 |
| 3.5 | 4.01 | 630.1 | 19.20 |
| 4.0 | 4.10 | 620.6 | 18.91 |
| 12.5 | 4.36 | 961.3 | 29.29 |

In Figure 5a,b, we present the time histories of different forms of energy at $\gamma = 12.5$ and $\gamma = 1.5$ respectively, to make a direct comparison between a small and a large distances. In Figure 5a, we observe that the potential energy of the bubble decreases during the process of collapse as the bubble shrinks, while the kinetic energy grows from zero, representing the accelerated process of the bubble collapse. As the bubble collapses to a singular point at 199 μs, the kinetic energy reaches its maximum value, while the potential energy drops to nearly zero value. During this whole period, the total energy is approximately equal to the initial bubble potential energy. In other words, the total mechanical energy remains approximately constant, or is conserved. During this process of bubble shrinkage, the potential energy is lost to the re-entrant jetting towards the bubble centre.

After the bubble vanishes, around 200 μs, a pressure wave is emitted and travels outwards, resulting in an intensive wave energy observed in Figure 5a. However, since we consider only limited forms of energy in the simulations and neglect thermal and acoustic damping mechanisms, which might be significant, the total energy is not conserved anymore. We understand that it is very challenging to quantify in detail any switch-over of these dominant damping mechanisms

in such violent bubble collapses. We leave it open to the future studies. To help understand this process more intuitively, we present in Figure 6 the flow fields for three time instants demonstrating the alternative dominance of different energy forms. In Figure 6a, at the early stage, the bubble potential energy dominates, while it is converted to kinetic energy when the bubble starts to collapse (see Figure 6b), and in Figure 6c the pressure wave is emitted with only the wave energy prominent in the energy budget.

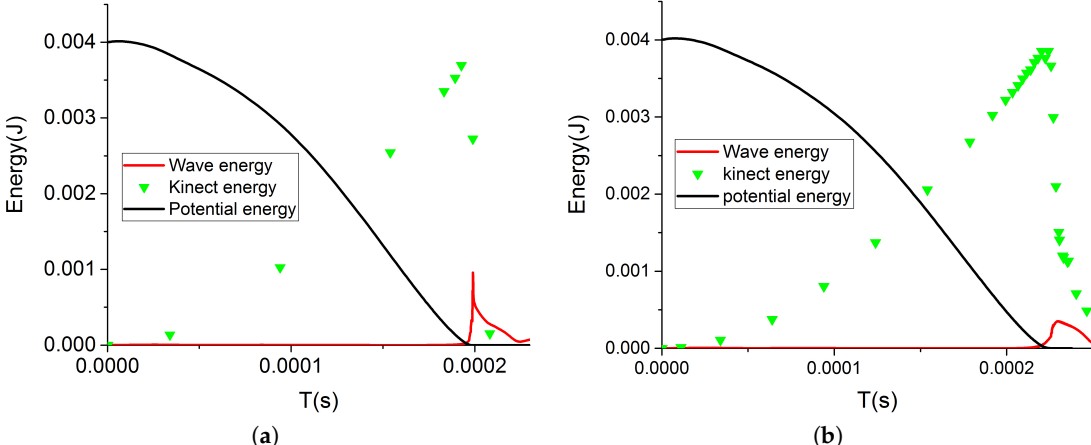

**Figure 5.** Time histories of different forms of energy in the flow domain during the collapse of a single bubble at (**a**) $\gamma = 12.5$ and (**b**) $\gamma = 1.5$. For both cases, $R_{max} = 2$ mm and $\Delta P = 101,325$ Pa.

For comparison, we present the time histories of different forms of energy at $\gamma = 1.5$ in Figure 5b, in which case the initial bubble is very close to the solid wall. Apparently, the changing trend of each form of energy is roughly the same with its large distance counterpart (see Figure 5a). They still differ in some aspects: first, the wave energy appears later than the large distance case due to the confinement of the solid wall, which delays the collapse of the bubble. Second, during the whole process, even after the kinetic energy drops, it is always higher than the wave energy, implying a different physical mechanism. We conjecture that the presence of a rigid boundary can lead to asymmetric collapse, which requires more time. In the process of asymmetric collapse, high speed micro jetting flows are induced towards the rigid boundary, the liquid can thus be pushed away along the wall instead of collapsing towards a singular point. Therefore, it is easy to understand that the kinetic energy dominates even after the bubble collapse.

To get a comprehensive understanding of the distance effects, in Figure 7 we present the variations of wave energy with time for various distances corresponding to Table 1. When the bubble is located far away from the rigid boundary, e.g., $\gamma = 12.5$, the wave energy reaches the maximum value of 0.95 mJ at $t = 1.85$ ms within a very short period of 0.1 ms, and with a very sharp peak. As the distance is reduced, the peak value of wave energy decreases accordingly, and the sharp peak is replaced by a broad distribution of the high wave energy. In specific, the peak value of wave energy for $\gamma = 1.5$ is around 0.3 mJ, only one-third of that for $\gamma = 12.5$. It is interesting to find that there are secondary peaks in the wave energy distributions at $\gamma = 2.5$–4.0. We believe that these secondary peaks are induced by the pressure waves reflected by the solid boundary, which are difficult to be distinguished as we further reduce the distance. The comparison between small and large distances suggests that the near-wall collapses generate less wave energy due to the non-spherical characteristics, but with a longer duration in the computational domain.

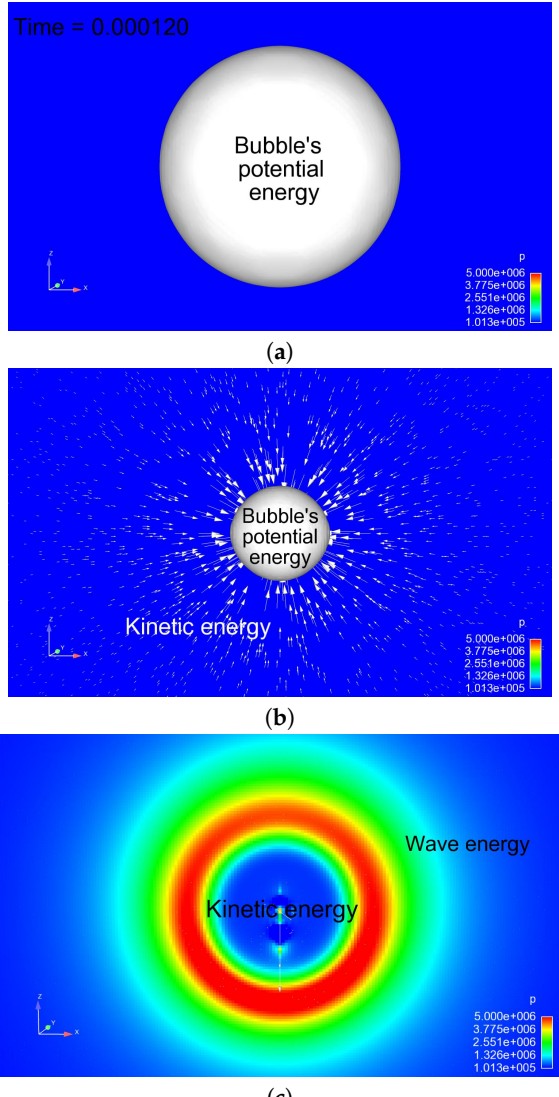

**Figure 6.** Flow fields with pressure contours for three time instants at $\gamma$ = 12.5, demonstrating different dominant forms of energy: (**a**) potential energy, (**b**) kinetic energy and (**c**) wave energy.

Figure 8 shows the variations of kinetic energy with time for various distances. Again, their overall trends are the same, as we have discussed in Figure 5, and they are consistent with the wave energy variations shown in Figure 7. There are also some slight differences among various distances. In short, the wall effects on kinetic energy can be concluded as: first, the drop of kinetic energy is delayed as the bubble is placed closer to the wall. Second, the sharp peak is replaced by broader distributions of the kinetic energy as the bubble stays closer to the wall. Both conclusions are consistent with the wave energy variations, as shown in Figure 7.

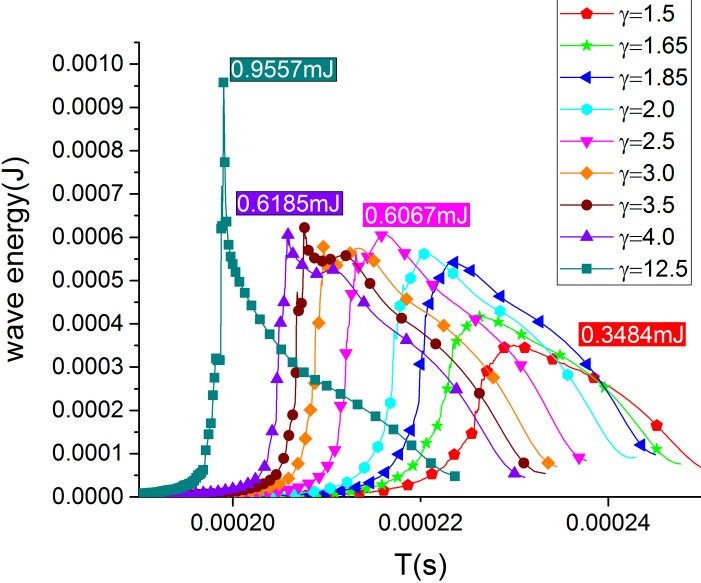

**Figure 7.** Wave energy variations in the flow fields at different values of $\gamma$.

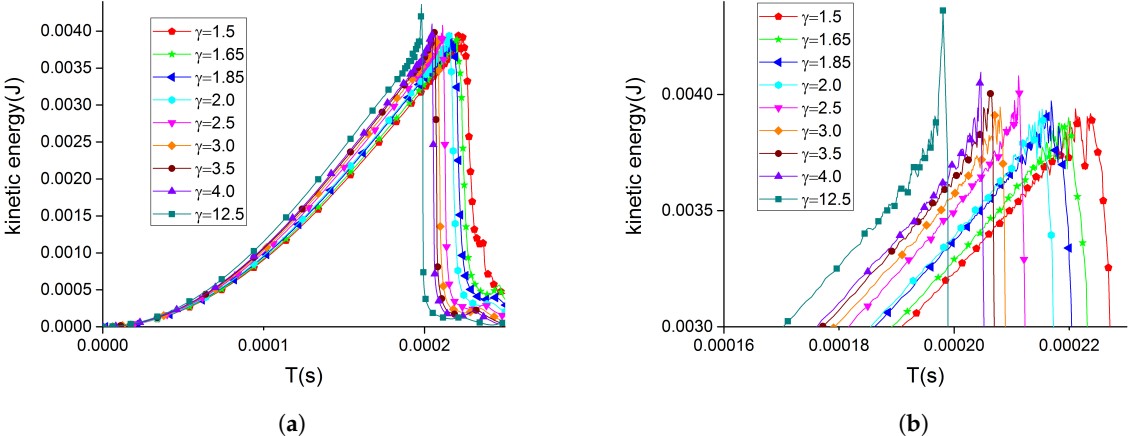

(**a**)　　　　　　　　　　　　　　　　　　　　(**b**)

**Figure 8.** (**a**) Kinetic energy variations in the flow fields at different values of $\gamma$, and the enlarged views around the peak regions are shown in (**b**).

### 3.2. Collapses of Multiple Bubbles

The asymmetric bubble collapse can be brought by placing a solid wall, as we have studied in the last section for a single bubble, it can also be caused by the presence of nearby bubbles. In this section, we study the collapses of multiple bubbles, or more specifically, 8 bubbles, which are placed in a cubic region with two layers, and 4 bubbles on each layer.

In Table 2, we present the peak values of both the kinetic and wave energies, as well as their conversion rates. The value of $E_k$ slightly grows with the distance $\gamma$, while $\eta$ decreases as $\gamma$ increases. In specific, the energy conversion rate at $\gamma = 1.5$ is approximately twice of that at $\gamma = 11.25$.

To examine the dynamic evolutions of individual bubbles, in Figures 9 and 10, we present the instantaneous shapes of the bubbles in three subsequent time instants. As shown in Figures 9a and 10a for the initial bubble configurations, the two cases represent respectively the small ($\gamma = 1.5$) and large ($\gamma = 11.5$) distances of the bubbles away from the solid wall. For the small distance, as shown in Figure 9, the wall effect leads to asymmetric collapses of the bubble cluster, with the bubbles in the upper layer collapsing faster than that in the lower layer, due to the solid wall below the lower layer. While at $\gamma = 12.5$, as shown in Figure 10, the asymmetry is brought only by the presence of nearby

bubbles, therefore, the collapses and the evolutions of bubbles are symmetric with respect to the symmetry plane between the two layers, which is a straightforward demonstration that the solid wall does not affect the bubble collapse markedly when they are spaced sufficiently far away. The two cases differ in the focal point. For the large distance case, the jetting flows from all bubbles point to the centre of the bubble cluster, while for the small distance case, the focal point shifts towards the solid wall.

**Table 2.** Energies and their conversion rates for multiple bubbles.

| $\gamma$ | $E_k$ (mJ) | $E_{wave}$ (mJ) | $\eta$ (%) |
|---|---|---|---|
| 1.5 | 27.96 | 2.97 | 11.31 |
| 2.0 | 28.65 | 2.96 | 11.27 |
| 3.75 | 29.91 | 2.09 | 7.96 |
| 6.25 | 29.28 | 2.03 | 7.73 |
| 11.25 | 30.04 | 1.40 | 5.33 |

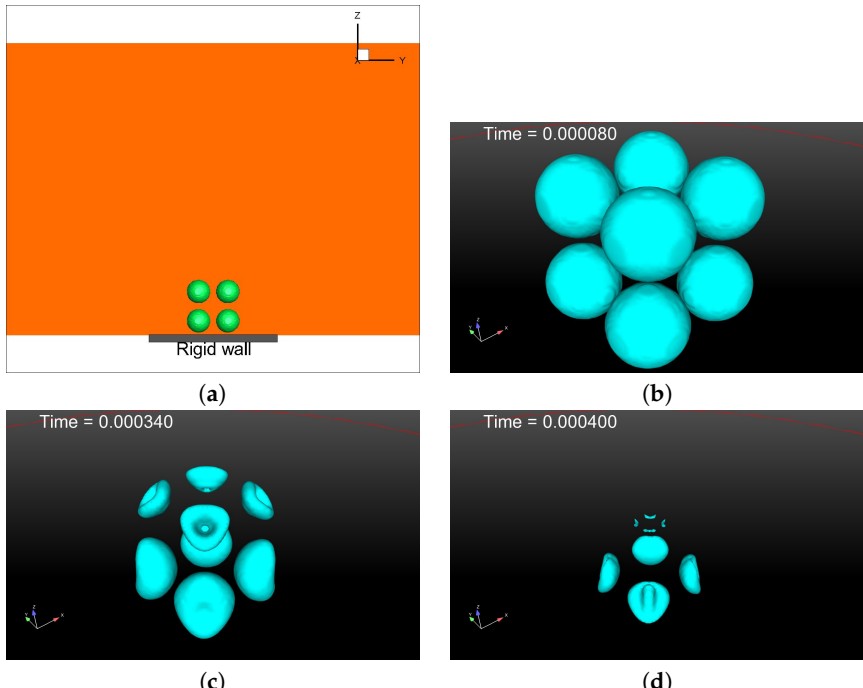

**Figure 9.** Deformations of multiple bubbles at $\gamma = 1.5$ for (**a**) the initial shapes (side view), (**b**) T = 80 μs, (**c**) T = 340 μs and (**d**) T = 400 μs.

Figure 11a presents the potential energy variations of multiple bubbles. The drops of potential energy slow down at the small distances, which is apparent, because the collapses of bubbles have been delayed due to the confinement of the solid wall. Comparing to the single bubble, for the multiple bubbles, the solid wall plays a similar role, as the bubbles are very close to the wall. However, the trend of wave energy conversion is quite different with that of the single bubble, which might be raised by the strongly nonlinear interactions between the bubbles. Similar comparisons can be made in the kinetic energy variations, as shown in Figures 8b and 11b.

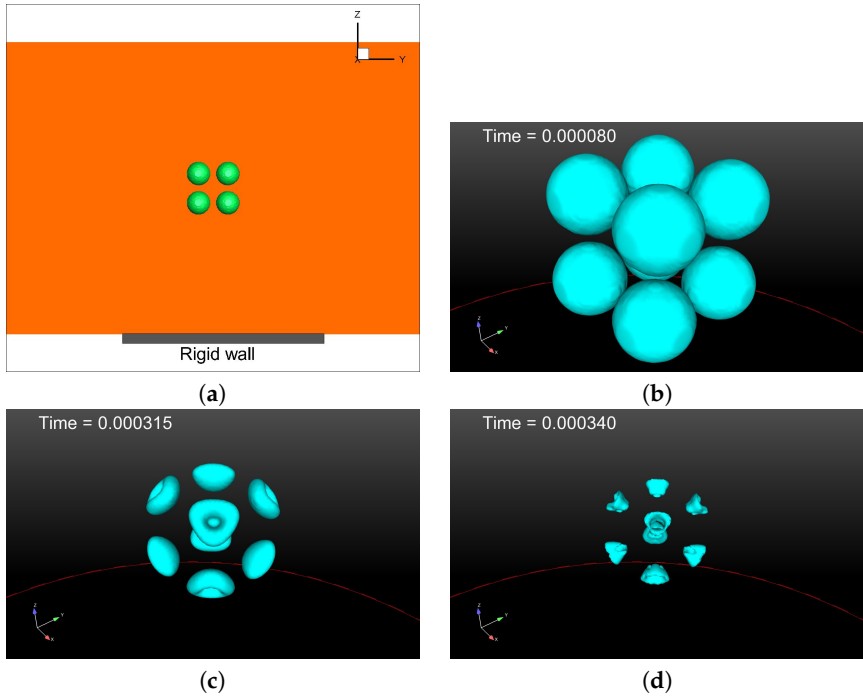

**Figure 10.** Deformations of multiple bubbles at $\gamma$ = 11.5 for (**a**) the initial shapes (side view), (**b**) T = 80 μs, (**c**) T = 315 μs and (**d**) T = 340 μs.

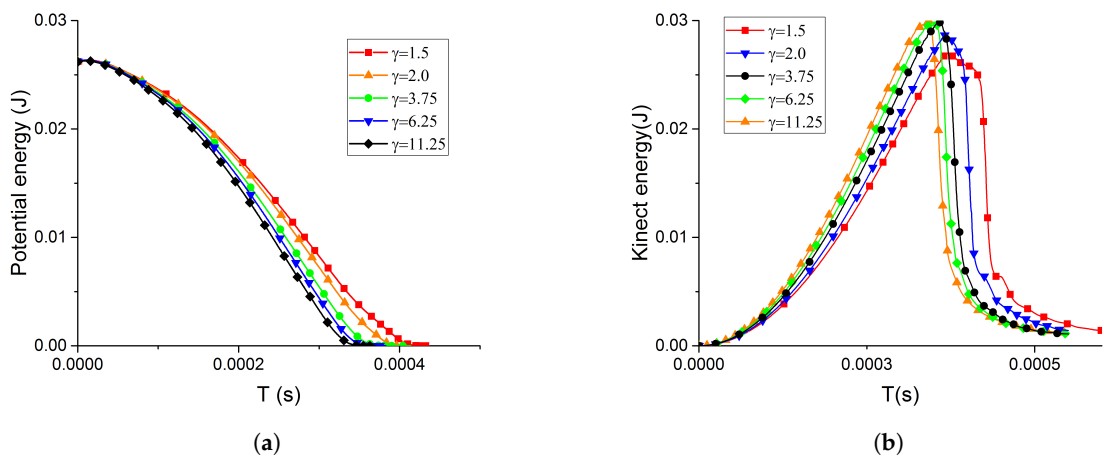

**Figure 11.** (**a**) Potential energy and (**b**) kinetic energy variations in the flow fields for the collapses of multiple bubbles.

The most remarkable difference between the single bubble and the multiple bubbles lies in the wave energy. Figure 12 presents the wave energy variations in the simulations of multiple bubbles. In contrast to that of the single bubble (see Figure 7), here, the ascending and descending branches of the wave energy variations are nearly symmetric with respect to the peak location. The peak values of wave energy for the small distance cases are higher than that of the large distance cases, resulting in the same behaviour for the energy conversion rate. This relationship between the peak wave energy is opposite to that of the single bubble.

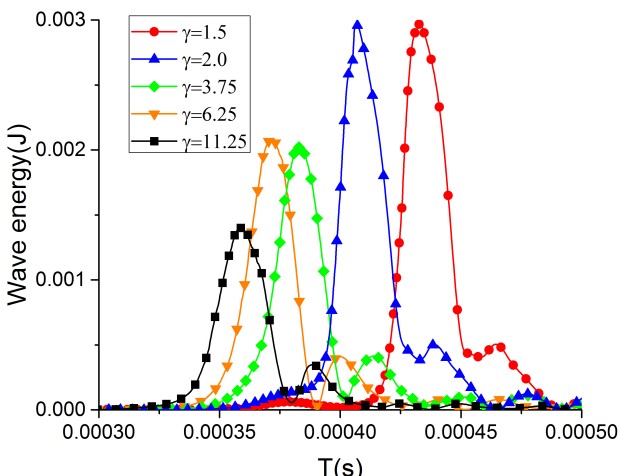

**Figure 12.** Variations of wave energy in the flow fields during the collapsed of multiple bubbles.

## 4. Conclusions

In the present work, we carry out numerical simulations to study the collapses for both a single bubble and a bubble cluster with 8 bubbles. Different forms of energy are evaluated to understand the underlying physical mechanisms of bubble collapse.

The first part concentrates on a single bubble collapsing near a solid wall. The intuitive physical rule tells that the energy should be conserved, which has indeed been observed during the first stage of collapse. As the bubble shrinks, the potential energy determined by the vapour volume converts into kinetic energy. Further, as the bubble vanishes, a part of kinetic energy starts to convert into other energy forms such as wave energy, or is lost due to the thermal and acoustic damping mechanisms, which have unfortunately not been considered in the current simulations. We report that a rigid boundary can affect the process of bubble collapse by deforming the bubble into non-spherical shapes, delaying the collapse, and consequently decreasing the conversion rate.

For a cluster consisting of 8 bubbles, their collapses are more complicated than the single bubble case, because the individual bubbles can be affected by both the solid wall and the surrounding bubbles, and both can break the symmetry of their geometric configurations. By examining the evolutions of individual bubbles during the collapse, we observe that when the bubbles are far away from the solid wall, the jetting flows from all bubbles point towards the cluster centre, while the focal point shifts towards the solid wall as the cluster is located very close to the wall. Moreover, we find that the collapse differentiates from the single bubble case mainly in the relationship between the bubble-wall distance and the wave energy, or equivalently the energy conversion rate.

**Author Contributions:** Conceptualization, J.Z. and L.Z.; Methodology, L.Z. and J.Z.; Investigation, J.Z.; Data Curation, J.Z.; Writing-Original Draft Preparation, J.Z. and L.Z.; Writing-Review and Editing, J.D.; Visualization, J.Z.; Funding Acquisition, L.Z. and J.D.

**Funding:** This research was funded by the National Natural Science Foundation of China (No. 11772298, No. 11272284) and the State Key Program of National Natural Science of China (No. 11332009).

**Conflicts of Interest:** The authors declare no conflict of interest.

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
