# Peer review of "Numerical Study of the Collapse of Multiple Bubbles and the Energy Conversion during Bubble Collapse"

_water, doi:10.3390/w11020247_

Round 1

Reviewer 1 Report

Note 1:

I would suggest writing out (at least once) the names of both the VOF method and PIV technique. Some readers may not immediately know what those acronyms mean.

Note 2:

Some variables from equations 1-3 and 8-11 are not thoroughly described (though some are described).

Note 3:

It seemed as if the potential energy, E_pot, that is discussed a fair amount should be accounted for in both Table 1 and Table 2. Readers can then see how E_pot and E_k change simultaneously as the gamma is changed.

Note 4:

Concerning gamma specifically, what happens as it approaches zero? Was there a specific reason that you had gamma range from 1.5-12.5 only? Does the efficiency continue to increase at larger gamma values, and if so, why stop at 12.5 (what is gamma's significance)? I think having a paragraph or so of text explaining your reasoning would assist readers in understanding your system geometry constraints.

Note 5:

On line 94, you write, "It is worth noting that ..." However, there is no subsequent explanation of what is important to note. This may confuse readers so possibly add some explanatory text.

Note 6:

In addition to Table 1, it may be beneficial for readers if you add another table which covers (for example gamma = 12.5) E_pot, E_k, and E_wave as a function of time. 

Note 7:

For figure 4, possibly add some labels which tell readers at what point the vapor bubble disappears. Lines 98-102 describe some of this (at 199 micro-seconds), but the figure suggests that the vapor bubble exists for another 50-60 micro-seconds.

Note 8:

For Figure 8, what is the inset figure showing? I was confused and could not find reference to it specifically in the text. If vital to the text, I would suggest pulling it out of Figure 8 and making it a separate figure.

Note 9:

Lines 146-147, was there a specific reason why wall distance ranges only go from 1.5-4.0? For single bubbles, you range from 1.5-12.5. Would there be any significant explanation for this difference? It may help readers again better understand the model constraints.

Note 10:

The comparison in Figure 11 would be beneficial to include in the single bubble section. This is a great explanatory figure.

Author Response

We thank the reviewer for a constructive and thorough review, which we believe has led to a substantially improved manuscript. First, the entire manuscript has been re-edited, all the grammatical errors have been corrected, and the English language has been polished. We believe that the revised manuscript is easy to read. And, to the reviewer's comments and suggestions, we make our replies one by one as follows, in which our replies are listed in red color.

Reviewer 2 Report

p.p1 {margin: 0.0px 0.0px 0.0px 0.0px; font: 12.0px 'Helvetica Neue'} p.p2 {margin: 0.0px 0.0px 0.0px 0.0px; font: 12.0px 'Helvetica Neue'; min-height: 14.0px} li.li1 {margin: 0.0px 0.0px 0.0px 0.0px; font: 12.0px 'Helvetica Neue'} ol.ol1 {list-style-type: decimal}

The manuscript examines the collapse of an individual bubble and of a cloud of bubbles situated near a rigid wall. The authors used a volume of fluid method to simulate the bubble dynamics. The results are interesting and new, and the conclusions are reasonable. I have no serious criticisms regarding the results presented in this manuscript and, in principle, it is suitable for publication. I do have, however, some comments that, I believe, would improve the presentation: 

Many times in the text: “Lanterborn” must be “Lauterborn”

Line 31: Define “L” and “Rmax”

Line 33: The maximum amplitude of the shock wave emitted during the collapse of a single bubble attached to a rigid wall is as much as 1.3 GPa (see, Experimental Thermal and Fluid Science, 32(2008), 1188-1191)

Give more details of the numerical method, such as discretization, method used for time integration, and initial conditions.

R0 in equation (8) must be Rmax (the maximum bubble radius)

English must be considerably improved. It was very difficult to read the present version of the manuscript. Almost everywhere in the text there are grammatical errors, misspellings, typographical errors, and faulty punctuation.

Author Response

We thank the reviewer for a constructive and thorough review, which we believe has led to a substantially improved manuscript. To the reviewer's comments and suggestions, we make our replies one by one as follows, in which our replies are listed in red color.

Round 2

Reviewer 1 Report

Thank you for all of the revisions and corrections.